# Association of Alcohol Types, Coffee, and Tea Intake with Risk of Dementia: Prospective Cohort Study of UK Biobank Participants

**DOI:** 10.3390/brainsci12030360

**Published:** 2022-03-08

**Authors:** Sylva Mareike Schaefer, Anna Kaiser, Inken Behrendt, Gerrit Eichner, Mathias Fasshauer

**Affiliations:** 1Institute of Nutritional Science, Justus-Liebig University of Giessen, 35390 Giessen, Germany; anna.kaiser@ernaehrung.uni-giessen.de (A.K.); inken.behrendt@ernaehrung.uni-giessen.de (I.B.); mathias.fasshauer@uni-giessen.de (M.F.); 2Mathematical Institute, Justus-Liebig University of Giessen, 35392 Giessen, Germany; gerrit.eichner@math.uni-giessen.de; 3Department of Internal Medicine (Endocrinology, Nephrology, and Rheumatology), University of Leipzig, 04103 Leipzig, Germany

**Keywords:** alcohol, body weight, coffee, dementia, non-wine, obesity, prospective cohort study, tea, wine

## Abstract

The prevalence of dementia is increasing globally and is linked to obesity and unfavorable dietary habits. The present study analyses the association of alcohol intake from wine and non-wine alcoholic beverages (non-wine) in g/d, as well as coffee and tea in cups/d, with incident dementia. Over 4.2 million person-years, 4270 dementia cases occurred in 351,436 UK Biobank participants. Hazard ratios (HRs) for incident dementia were defined with Cox proportional hazard regression models in which beverage intake was fitted as penalized cubic splines. Wine intake showed a significant U-shaped association with the lowest risk for incident dementia (nadir) ranging from 21 to 23 g alcohol/d in all participants and in males. In contrast, non-wine consumption was significantly and dose-dependently associated with incident dementia, and the nadir was found at 0 g alcohol/d. Coffee consumption was not related to dementia risk, while moderate-to-high tea intake was negatively associated with incident dementia. Taken together, the current study shows on a population level that moderate consumption of wine and moderate-to-high tea intake is associated with a decreased risk of incident dementia. In contrast, non-wine is positively related to dementia risk in a linear fashion, and no clear association is found for coffee.

## 1. Introduction

Dementia is a condition characterized by a decline in cognitive function going beyond the common effects of biological aging [1]. Currently, there are more than 55 million people living with dementia worldwide, and almost 10 million new cases occur every year [1]. The risk of developing dementia is positively associated with overweight and obesity at younger but not older ages [2]. Despite intensive research throughout the recent decades, no effective pharmacological treatment for the main dementia types has yet been developed [3].

Lifestyle interventions, such as dietary changes, are a feasible approach not only to combat obesity [4] but also to delay the deterioration of cognitive function from mild cognitive impairment to dementia [5]. Thus, daily consumption of unprocessed or minimally processed plant-based food, such as fruits, vegetables, olive oil, and nuts, contributes to healthy body weight [6] and has neuroprotective effects [7,8]. Correspondingly, adhering to the Mediterranean diet is associated with lower body weight [9], as well as with a decreased risk of cognitive impairment and dementia [7,10,11]. Furthermore, sugar-sweetened beverages increase the risk of both obesity and dementia [12,13]. In contrast, results for other beverages, including alcoholic drinks, coffee, and tea with incident obesity and dementia, have been less clear. Both alcoholic beverages and caffeine exert pharmacological effects on the brain. Alcoholic beverages increase microglial activation, neuroinflammation, and neuronal cell death [14]. In contrast, caffeine exerts neuroprotective effects on the brain, such as a reduction in β-amyloid peptide production via secretase suppression [15], as well as a decrease in neuroinflammation through lowering of extracellular calcium, glutamate release, and activation of microglia [16]. Therefore, it is of importance to evaluate the impact of alcohol, coffee, and tea on cognitive function.

Alcohol intake is measured in categories such as no, light (≤12 g alcohol/d), moderate (>12 to ≤24 g alcohol/d), and heavy (>24 g alcohol/d) [17]. The National Health Service recommends a restriction of alcohol intake to less than 16 g/d for both sexes to maintain good health [18]. However, guidelines do not distinguish between sources of alcohol, i.e., wine versus all non-wine alcoholic beverages combined (non-wine), including beer and spirits. Studies have suggested that the association between wine and incident obesity [19] and diabetes mellitus [20] is different as compared to non-wine. Furthermore, a negative association between wine and incident dementia has been described [17,21], while a positive [17] or no [21,22,23] association for beer and spirits exists. Moreover, the association of wine consumption with cognitive decline might depend on sex [24].

Coffee consumption is measured in cups/d with moderate intake between three and five cups/d [25]. Results for coffee intake and decline of cognitive function have been discordant with findings suggesting increased odds of dementia [26], no association [27], or a lower dementia risk [28,29]. Furthermore, the protective effects of coffee were more pronounced in females as compared to males [28]. Additionally, lifetime exposure to caffeine reduces the risk for incident dementia for both men and women, also with a greater effect in women [30].

Tea intake is measured using the following categories: no, light (zero to two cups/d), moderate (three to four cups/d), and high (more than five cups/d) [31]. In contrast to coffee, no specific recommendation for optimal tea intake exists. Several studies have shown a negative association between tea intake and cognitive impairment [29,32,33], Alzheimer’s disease [33], and dementia [33,34]. No sex-dependent differences regarding the association between tea intake and dementia risk were observed [31,35].

To the best of our knowledge, the four beverage types, i.e., wine, non-wine, coffee, and tea, have not been assessed within a large prospective cohort study with mutual adjustment. Furthermore, previous models often include alcohol, coffee, and tea consumption as linear or discretized ordinal predictors. Moreover, wine consumption has not been contrasted with non-wine intake regarding incident dementia, and effects of abstainer bias on potential protective findings of alcohol consumption [36] have not been assessed systematically.

To address these limitations, we analyzed the associations between wine, non-wine, coffee, and tea intake and proportional hazards for incident dementia in 351,436 UK Biobank participants using penalized cubic splines. Wine was contrasted with non-wine since only wine contains substances such as resveratrol [37,38] that may have neuroprotective effects, and median intake levels between alcoholic beverage groups should be similar to improve the comparability of the data. We hypothesized that non-linear relationships exist between the four beverages and the risk for incident dementia, as well as that intake levels linked to the lowest hazards depend on the analyzed beverage.

## 2. Materials and Methods

### 2.1. Study and Participants

The UK Biobank study is a multicentre, prospective cohort study and its study design is described in more detail at https://www.ukbiobank.ac.uk [39], accessed on 7 February 2022. Between 2006 and 2010, more than 500,000 participants aged 38 to 73 years had their baseline assessment throughout the UK as described recently [40]. To all analyses, we applied the following five exclusion criteria (ec): (1) pre-existing dementia at baseline and incident dementia within 2 years after baseline (landmark analysis), (2) missing smoking status, (3) missing socioeconomic status (i.e., annual household income (AHI), ethnicity, highest qualification and/or overall health rating (OHR)), (4) missing percentage body fat, (5) either missing information on beverage intake or being in the upper 0.1% of alcohol, coffee, or tea consumption.

To obtain the primary cohort, all non-alcohol drinkers (present alcohol intake of 0 g alcohol/d) were removed from the analysis to exclude participants not drinking alcohol due to health issues (primary cohort; *n* = 351,436; Appendix A). The group of non-drinkers was comprised of lifetime abstainers and former drinkers, the latter drinking alcohol in the past but not in the present. Two sets of sensitivity analyses (S1 and S2) were performed in addition to the primary analyses. Firstly, only former drinkers but not lifetime abstainers were excluded from the analysis in addition to ec1 to ec5 (cohort S1; *n* = 371,153). Former drinker bias may contribute to the seemingly lower morbidity of moderate drinkers since the category of non-drinkers includes ex-drinkers who quit alcohol consumption because of poor health [41,42]. Secondly, all non-drinkers were included in the analyses, i.e., only ec1 to ec5 were applied (cohort S2; *n* = 395,893). Hence, this second set of analyses did not control for the abovementioned potential health issues of participants not drinking alcohol. All analyses on the influence of wine, non-wine, coffee, and tea on dementia incidence were conducted in the primary cohort, as well as in cohorts S1 and S2. All participants provided their written informed consent before inclusion in the study, which was approved by the North West Multicentre Research Ethics Committee [39].

### 2.2. Exposure Assessment

The assessment of alcohol intake from wine and non-wine, as well as consumption of coffee and tea, was performed as described recently by our group [43]. In brief, wine intake was defined as red wine and champagne plus white wine consumption, while all other categories of alcoholic beverages, i.e., beer plus cider, spirits, fortified wine, and other alcoholic drinks, were defined as non-wine. For all alcoholic beverages, an alcohol content of 10 g per portion was assumed, except for a pint of beer for which content of 20 g alcohol was defined. Coffee and tea intake was documented in cups/d. Participants were excluded if the extent of alcohol, coffee, or tea intake was not specified or questions concerning beverage intake were answered with “do not know” or “prefer not to answer”.

### 2.3. Outcome Assessment

The UK Biobank provides morbidity data as the earliest record date and respective health outcome defined by three-character International Statistical Classification of Diseases and Related Health Problems, Tenth Revision (ICD-10) codes [44]. Record sources were self-report at baseline assessment, as well as inpatient hospital, primary care, and death record data [44]. The primary outcome of the present study was incident dementia defined as ICD-10 codes F00, F01, F02, F03, G30, and G31. Follow-up time was calculated by subtracting the date of the baseline assessment from the date of first dementia occurrence, loss-to-follow-up, death, or censoring (i.e., 31 March 2021), whichever came first. In the case of several dementia diagnoses for a patient, the lowest duration to diagnosis was used. All analyses were performed in the total cohort and by sex.

### 2.4. Statistical Analyses

Data were imported, processed, analysed, and graphically displayed with R version 4.0.5 [45] and the packages readxl [46], tidyverse [47], venn [48], skimr [49], and survival [50,51,52]. Baseline characteristics of UK Biobank participants depending on sex were compared using Chi-squared test for categorical parameters and Mann–Whitney U test for continuous variables. Wine, non-wine, coffee, and tea were mutually adjusted within Cox proportional hazard regression models and included as penalized cubic splines as described recently [43]. The lowest estimated hazard ratio (HR) over the range from zero to the 99%-quantile of beverage consumption was defined as the nadir and set to 1. For all morbidity analyses, HRs with pointwise 95% confidence intervals (CIs) are depicted, and HR^0^ reflects HR in non-consumers relative to the HR at the nadir. No further interpretation of the nadir or HR^0^ was performed if both linear and non-linear *p*-values were non-significant. All covariates violating the proportional-hazard assumption after Holm-adjustment for multiple testing were stratified in the final models. All models were adjusted for sex (all participants only), age (quartiles), AHI (<18, 18 to <31, 31 to <52, 52 to <100, ≥100 k£, and unknown), ethnicity (White, group combined of Mixed, Asian, Black, Chinese, and Other), highest qualification (none of the below, national exams at age 16 years, vocational qualifications, optional national exams at ages 17–18 years, professional, College or University), OHR (poor, fair, good, excellent), physical activity (PA: metabolic equivalent of task-min per week: <1000, 1000 to <2000, 2000 to <4000, ≥4000, and unknown), percentage body fat (quartiles), and smoking status (never, previous, current). The following sensitivity analyses were run in the primary cohort: ICD-10 code G31 was excluded from the analysis as an endpoint since it is more of a pathological diagnosis rather than a functional one. Furthermore, age and percentage body fat were included in the Cox proportional hazard regression models as continuous instead of categorical variables, with the latter being square rooted to better approximate a normal distribution. A *p*-value of < 0.05 was considered statistically significant in all analyses.

## 3. Results

### 3.1. Baseline Characteristics and Dementia Cases in UK Biobank Participants

Baseline data of the UK Biobank study population in total and depending on sex are presented in Table 1. Median (Quartile (Q) 1, Q3) age of the study population was 58 (50, 63) years, with 50.7% of participants being female (Table 1). Median (Q1, Q3) consumption was 5.7 (1.4, 11.4) g alcohol/d from wine, 4.3 (0.0, 12.9) g alcohol/d from non-wine, 2.0 (0.5, 3.0) cups/d coffee, and 3.0 (1.0, 5.0) cups/d tea (Table 1).

Wine intake was similar in both sexes (females 5.7 (2.9, 11.4); males 5.7 (0.7, 11.4) g alcohol/d); however, statistical significance was reached due to a high sample size (*p* < 0.0001) (Table 1). In contrast, non-wine intake was much lower in female (1.4 (0.0, 4.3) g alcohol/d) compared to male (11.4 (4.3, 22.9) g alcohol/d) participants (*p* < 0.0001) (Table 1). Coffee (females 2.0 (0.5, 3.0); males 2.0 (1.0, 3.0) cups/d) and tea (females and males 3.0 (1.0, 5.0)) intake was comparable between both sexes; however, statistical significance was again reached due to a high number of participants (*p* < 0.0001 for coffee; *p* < 0.001 for tea) (Table 1).

During 4.2 million person-years and a median (Q1, Q3) follow-up of 12.0 (11.3, 12.7) years, a total of 4270 incident dementia cases occurred, i.e., 1704 and 2566 cases in females and males, respectively.

### 3.2. Beverage Intake and Dementia Risk

**Wine intake:** In all participants and in males, a significant U-shaped association between wine intake and HR for incident dementia was detected (Figure 1a,c). The nadir was observed at 21 and 23 g alcohol/d in all participants and males (Figure 1a,c). HR^0^ was 1.19 (1.13, 1.24) in all participants and 1.17 (1.11, 1.23) in males (Figure 1a,c). A significant U-shaped relation between wine intake and incident dementia was also seen in cohorts S1 and S2 with the nadir between 17 and 23 g alcohol/d in all participants, as well as in both sexes separately (Appendix A). Furthermore, an increased HR^0^ was detected in cohorts S1 and S2, with a most pronounced elevation seen in the latter. Compared to the HR^0^ in the primary cohort (Figure 1b,c), HR^0^ in cohort S2 was higher in both females (1.40 (1.32, 1.49); Appendix A) and males (1.28 (1.22, 1.34); Appendix A). The nadir was at 29 g alcohol/d in all participants if ICD-10 code G31 was removed from the analysis (Appendix A). No relevant alteration was observed if age and percentage body fat were included as continuous covariates as compared to the model, including them as categorical parameters (Appendix A).

**Non-wine intake:** In all participants and both sexes, a significant positive dose-dependent association between non-wine consumption and HR for incident dementia was seen with the nadir at 0 g alcohol/d (Figure 2). The shape of the association was comparable but slightly more flattened in cohorts S1 and S2 compared to the primary cohort (Appendix A). The nadir was detected at higher consumption levels with a most pronounced shift towards higher values in cohort S2 (Appendix A). Furthermore, a significant HR^0^ was only seen in cohort S2 in all participants (1.08 (1.05, 1.11); Appendix A) and males (1.12 (1.06, 1.18); Appendix A). The association between non-wine intake and dementia risk was somewhat blunted but still significant in all participants if ICD-10 code G31 was removed from the analysis (Appendix A). Inclusion of age and percentage body fat as continuous instead of categorical covariates did not significantly affect the association between non-wine consumption and incident dementia (Appendix A).

**Coffee intake:** Coffee consumption was not significantly associated with dementia risk in all participants, females and males in the primary cohort (Figure 3). The results remained virtually unchanged in cohorts S1 and S2 (data not shown), as well as in the analysis, including age and percentage body fat as continuous covariates in all participants (Appendix A). A significant dose-dependent association beyond three cups/d coffee with dementia risk was observed if ICD-10 code G31 was removed from the analysis (Appendix A).

**Tea intake:** In all participants and males, a significant U-shaped association was seen between tea intake and incident dementia (Figure 4a,c). HR^0^ was significantly increased in all participants (1.23 (1.15, 1.32); Figure 4a) and males (1.31 (1.21, 1.43); Figure 4c). The nadir was at six cups/d for all participants and seven cups/d for males (Figure 4a,c). Similar findings were observed in all participants in cohorts S1 and S2 (data not shown), after removal of ICD-10 code G31 from the analysis (Appendix A), as well as after inclusion of age and percentage body fat as continuous instead of categorical parameters (Appendix A).

## 4. Discussion

The current study elucidates for the first time how wine, non-wine, coffee, and tea intake included as continuous non-linear predictors and mutually adjusted is associated with incident dementia. Furthermore, the impact of the abstainer bias on potential protective effects of alcohol consumption is systematically assessed.

For wine intake, we show a significant U-shaped association with incident dementia for all participants (primary cohort), and HR^0^ is significantly increased compared to the nadir at 21 g alcohol/d. Compared to our current findings, varying levels of wine consumption are associated with the lowest risk of dementia, i.e., an intake of up to three daily servings [21], three to four glasses [53], or a nadir of 6 g alcohol/d [17], while no association is found in another report [23]. The current study shows a significant U-shaped association between wine consumption and incident dementia for male participants with the nadir at 23 g alcohol/d. A study from Denmark shows comparable results with monthly and weekly wine consumption linked with a decreased dementia risk in both sexes [54]. Red wine intake is negatively associated with the incidence of Alzheimer’s disease in men only, while a positive association is seen in women [24]. The potential neuroprotective effect of wine might be caused by natural ingredients of wine not present in non-wine beverages, such as the phenolic substance resveratrol found in the epidermis of red grapes [38]. In rats, resveratrol inhibits the apoptosis pathway and exerts anti-oxidative effects [38]. Furthermore, resveratrol derivatives, such as stilbenoids, modulate multiple mechanisms of the neurodegenerative disease pathology, including inhibition of β-secretase, reactive oxygen intermediates generation, and β-amyloid peptide aggregation [37].

To the best of our knowledge, our study is the first to define the relationship between non-wine consumption and incident dementia. Contrary to wine intake, a significant positive dose-dependent association exists for non-wine consumption with the nadir at 0 g alcohol/d for all participants and females and males. Within non-wine beverages, several studies have examined the association of beer and/or spirits with incident dementia with conflicting results. In accordance with our findings, an elevated risk for highest versus lowest category of beer consumption is found in a large meta-analysis of prospective studies [17]. A positive association between spirits consumption and incident dementia is observed in a report in women [55]. Monthly beer but not spirits consumption is associated with higher dementia risk in another study [54]. In contrast, no significant association is found between intake of liquor and beer on the one hand and Alzheimer’s disease [21,56] or cognitive function [57] on the other hand. Together, these data suggest that even at light-to-moderate intake levels, non-wine is not consistently associated with a decreased risk of dementia and might even be related to heightened risk in contrast to wine.

To the best of our knowledge, our study is the first to systematically assess the impact of former drinkers and lifetime abstainers on the association between wine and non-wine consumption on one hand and incident dementia on the other hand. We show that HR^0^ is higher after including all non-drinkers (cohort S2) as compared to the primary cohort for both wine and non-wine. In addition, the association of non-wine with incident dementia is more J-shaped, and a shift of the nadir towards higher levels in cohort S2 is detected as compared to the primary cohort. In contrast to our primary analyses, studies on wine and non-wine alcoholic beverages [17,21,23,24,53,55,56,57] do not exclude non-drinkers from the reference group. The category of non-drinkers might include former drinkers and never-drinkers who stopped or never initiated drinking because of bad health or emerging cognitive decline [36]. If these non-drinkers are included in the reference category, an apparent protective effect of light-to-moderate drinking may only stem from the already increased dementia risk among former and never drinkers [58]. Thus, it is important to exclude non-drinkers from the analysis since it could lessen or erase any observed protective effects and change the shape of the association [36,58].

For coffee intake, we show no significant association with dementia risk. Consistent with the present findings, no association between coffee consumption and risk of Alzheimer’s disease and incident dementia is shown in a large meta-analysis of prospective studies [59]. Similarly, no reduction in the risk of cognitive decline is found in a longitudinal study [60]. Two previous studies have assessed the association between coffee intake and dementia risk in UK Biobank participants using different analytical approaches [26,34]. Based on hospital inpatient records and cubic splines, Zhang and co-workers demonstrate convincingly that the risk of dementia is lowest at two to three cups/d coffee [34]. Pham et al. show that the coffee intake category >6 as compared to the 1–2 cups/d category is associated with 53% higher odds of dementia [26]. Differences in study results may be well explained by differences in model adjustments, exclusion criteria, and follow-up period. Interestingly, light-to-moderate coffee consumption is linked to a reduced risk of any cognitive deficits or dementia in a recent meta-analysis of 29 prospective studies [29]. Taking the current findings and published studies into account, a light-to-moderate amount of coffee consumption is not positively related to dementia risk, and a minor negative association remains possible.

For tea consumption, we show a significant U-shaped association with incident dementia in all participants and males. HR^0^ is significantly increased compared to the nadir at six (all participants) and seven (males) cups/d. Zhang and co-workers show convincingly that the HR for incident dementia is lowest at three to five cups/d in UK Biobank participants using hospital inpatient records and cubic splines [34]. A negative association between tea consumption and cognitive deficits was observed in a recent meta-analysis [29]. Few studies have compared green and black tea separately. Interestingly, an inverse association for incident dementia with green tea but not black tea consumption is detected in two independent cohorts [31,61]. Unfortunately, the type of tea has not been recorded for all participants during the baseline assessment of the UK Biobank; however, the UK has one of the highest per capita consumption levels of black tea worldwide [62]. The current findings and published results are in agreement with the hypothesis that overall tea consumption is associated with a lower risk of dementia. Potential differences between green and black tea need to be systematically assessed in future studies.

The current study has several strengths, such as a prospective cohort design, a large number of well-characterized participants, a long follow-up period, as well as a wide range of beverage intake. However, some limitations must be considered: No information is available concerning the consumption of other important beverages, e.g., sugar-sweetened beverages and milk-based drinks, as well as the type of tea. Furthermore, only three-character ICD-10 codes are available for morbidity data. Therefore, the diagnosis F10.6 is not included as a dementia diagnosis in the current study since it could not be distinguished from other common codes within F10, e.g., F10.1 and F10.2. Moreover, the groups of participants not drinking coffee (*n* = 66,124) and tea (*n* = 48,568) are too small to define associations of coffee and tea intake with dementia risk in participants not drinking tea and coffee, respectively. Residual confounding, measurement errors in the assessment of the exposure variables, and a “healthy volunteer” selection bias [63] are further limitations.

Taken together, the current study shows on a population level that moderate consumption of wine and moderate-to-high tea intake is associated with a decreased risk of incident dementia. In contrast, non-wine is positively related to dementia risk in a linear fashion, and no clear association is found for coffee. Further prospective studies should elucidate potential differences between green and black tea.

## Figures and Tables

**Figure 1 brainsci-12-00360-f001:**
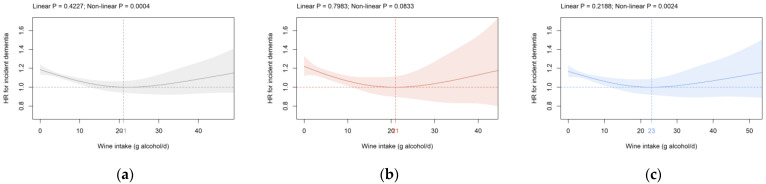
Association of wine intake (g alcohol/d) with dementia risk in the primary cohort in (**a**) all participants, (**b**) females, and (**c**) males. Data are adjusted for sex (all participants only), age, AHI, ethnicity, highest qualification, OHR, PA, percentage body fat, and smoking status. Wine, non-wine, coffee, and tea intake are mutually adjusted (e.g., wine intake is additionally adjusted for non-wine, coffee, and tea intake). The nadir is indicated in grey (total cohort), red (female), and blue (male). AHI: Annual household income; HR: Hazard ratio; OHR: Overall health rating; PA: Physical activity.

**Figure 2 brainsci-12-00360-f002:**
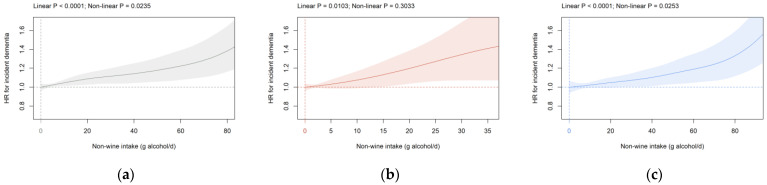
Association of non-wine intake (g alcohol/d) with dementia risk in the primary cohort in (**a**) all participants, (**b**) females, and (**c**) males. Data are adjusted and presented as indicated in Figure 1.

**Figure 3 brainsci-12-00360-f003:**
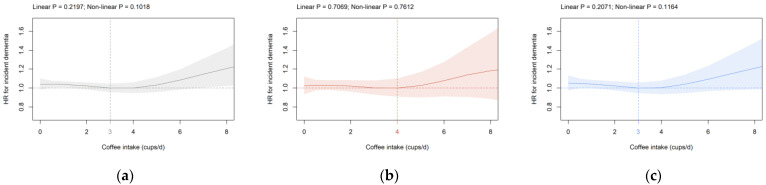
Association of coffee intake (cups/d) with dementia risk in the primary cohort in (**a**) all participants, (**b**) females, and (**c**) males. Data are adjusted and presented as indicated in Figure 1.

**Figure 4 brainsci-12-00360-f004:**
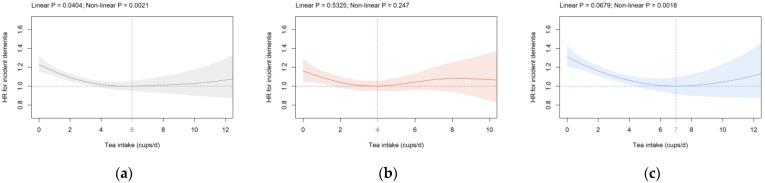
Association of tea intake (cups/d) with dementia risk in the primary cohort in (**a**) all participants, (**b**) females, and (**c**) males. Data are adjusted and presented as indicated in Figure 1.

**Table 1 brainsci-12-00360-t001:** Baseline characteristics of the UK Biobank cohort depending on sex ^1^.

Parameter	All (*n* = 351,436)	Female (*n* = 178,389)	Male ^#^ (*n* = 173,047)
**Wine intake (g/d)**	5.7 (1.4, 11.4)	5.7 (2.9, 11.4)	5.7 (0.7, 11.4)
**Non-wine intake (g/d)**	4.3 (0.0, 12.9)	1.4 (0.0, 4.3)	11.4 (4.3, 22.9)
**Coffee intake (cups/d)**	2.0 (0.5, 3.0)	2.0 (0.5, 3.0)	2.0 (1.0, 3.0)
**Tea intake (cups/d)**	3.0 (1.0, 5.0)	3.0 (1.0, 5.0)	3.0 (1.0, 5.0)
**Age (years)**	58 (50, 63)	57 (50, 63)	58 (50, 63)
**AHI (k£)**			
<18	55,960 (15.9)	29,328 (16.4)	26,632 (15.4)
- 18 to <31	76,591 (21.8)	39,078 (21.9)	37,513 (21.7)
- 31 to <52	85,290 (24.3)	41,302 (23.2)	43,988 (25.4)
- 52 to <100	71,810 (20.4)	33,190 (18.6)	38,620 (22.3)
- ≥100 - Unknown	20,161 (5.7) 41,624 (11.8)	9121 (5.1) 26,370 (14.8)	11,040 (6.4) 15,254 (8.8)
**Ethnicity**			
- White	340,001 (96.7)	172,819 (96.9)	167,182 (96.6)
- Mixed, Asian, Black, Chinese, Other	11,435 (3.3)	5570 (3.1)	5865 (3.4)
**Highest Qualification**			
- None of the below	50,416 (14.6)	24,189 (13.6)	26,227 (15.2)
- National exams at age 16 years	58,119 (16.5)	34,787 (19.5)	23,332 (13.5)
- Vocational qualifications	38,492 (11.0)	14,660 (8.2)	23,832 (13.8)
- Optional national exams at ages 17–18 years	26,619 (7.6)	14,469 (8.1)	12,150 (7.0)
- Professional	52,545 (15.0)	27,559 (15.4)	24,986 (14.4)
- College or University	125,245 (35.6)	62,725 (35.2)	62,520 (36.1)
**OHR**			
- Poor	10,478 (3.0)	4229 (2.4)	6249 (3.8)
- Fair	66,120 (18.8)	29,189 (16.4)	36,931 (21.3)
- Good	211,398 (60.2)	110,514 (62.0)	100,884 (58.3)
- Excellent	63,440 (18.1)	34,457 (19.3)	28,983 (16.7)
**PA (MET-min/week)**	1800 (845, 3546)	1764 (838, 3375)	1857 (848, 3714)
**Percentage body fat**	30.2 (24.7, 36.7)	36.2 (31.6, 40.7)	25.3 (21.5, 28.9)
**Smoking status**			
- Never	185,929 (52.9)	102,189 (57.3)	83,740 (48.4)
- Previous	130,726 (37.2)	61,360 (34.4)	69,366 (40.1)
- Current	34,781 (9.9)	14,840 (8.3)	29,941 (11.5)

^1^ Categorical variables are presented as number (percentage) and continuous variables as median (Q1, Q3); AHI: Annual household income; MET: Metabolic equivalent of task; OHR: Overall health rating; PA: Physical activity; Q: Quartile, ^#^ indicates *p*-value < 0.001 as assessed by Chi-squared test for categorical variables and Mann–Whitney U test for continuous variables.

## Data Availability

Data supporting the results of this study are available from UK Biobank, but restrictions apply to the availability of these data, which were used under license for Application 53438, and so are not publicly available.

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
