# Peer review of "Association of Alcohol Types, Coffee, and Tea Intake with Risk of Dementia: Prospective Cohort Study of UK Biobank Participants"

_brainsci, 2022, doi:10.3390/brainsci12030360_

Round 1
Reviewer 1 Report
Schaefer et al. address in this paper the association of wine and non-wine alcoholic consumption, as well as coffee and tea drinking, with the risk of incident dementia.
This work is based on data from a large and well-characterized cohort and is strengthened by an impressively robust and accurate statistical modelling. The interdependent adjustment, the use of non-linear model, and an accurate control for the abstainer bias are strengths of the study and remarkably contribute to a better knowledge about the topic that has been addressed before, as exhaustively described by the authors in the introduction.
However, the conclusions that are drawn do not seem to agree with the presented results. Both in the abstract and in the conclusions (ll. 26-27, 329-330) the authors claim that "light to moderate consumption of wine and tea is associated with a decreased risk of incident dementia". Figure 1(a) shows that people consuming up to around 12 g/d of wine-alcohol (i.e., more than one standard drinking unit a day) have higher risk of incident dementia. The nadir for the HR is set on 23 g/d, which is way beyond the higher limit of recommended alcohol intake, as the authors also report (l. 54). So, it seems instead that light wine drinking is detrimental in terms of incident dementia. Also, in figure 2(a) consumption of up to 3 cups/day of tea is associated with higher risk of incident dementia. The nadir is reached at 5 cups/day and the risk is not significantly higher up to 12 cups/day. Even if there is not standardized classification of tea drinking, I would interpret this as moderate-to-heavy tea drinking (>5 cups/day) has protective effects against dementia.
This concern must be appropriately addressed in order not to invalidate the overall high quality of the paper.
Additional concerns:
- “In agreement with our current finding, light to moderate wine consumption is associated with a lower risk of dementia in most reports” (ll. 248-249). The referred papers show very varying levels of wine intake as nadir risk for dementia. This should be discussed more diffusely.
- No intervals are given according to categorization of drinking into light, moderate, and heavy (ll. 53). Reference must be provided.
- In the primary outcome, ICD-10 diagnosis F10 (alcoholic dementia) is not included, while G31 (degenerative CNS disease), which is more of a pathological diagnosis rather than a functional one, is included. Can the authors argue for this?
- Why were continuous confounding variables, such as age, AHI, percentage body fat, divided into quartiles and treated as categorical (ll. 163-168)?
- Important risk factors for dementia are low educational attainment, cognitive inactivity, and social isolation. These variables, and especially educational level, should be considered as confounding factors in the statistical modelling.
- Data from table one is presented in ll. 180-184 with comparisons ("similar", "lower", "comparable"), but no statistical results are given. This should be calculated and added to the table.
Minor concerns:
- Ll. 38-39: lifestyle interventions are fundamental irrespectively of the availability of pharmacological treatments.
- Ll. 48-50: “alcoholic beverages and caffeine may exert pharmacological effects on the brain”: what is a pharmacological effect? Maybe CNS-stimulating or CNS-depressive effects is intended?
- Ll. 66: what is “steady” caffein intake? Please clarify.
- Supplementary figure 1: in the Eulero-Venn graph, the labels should define what is inside the circle, which is not the case in its present form (e.g., "0 g/dl alcohol", I guess it refers to what is outside the circle).
Reviewer 2 Report
Review of manuscript entilted: “Association of alcohol types, coffee, and tea intake with risk of dementia: prospective cohort study of UK Biobank participants” authored by Sylva Mareike Schaefer, Anna Kaiser, Inken Behrendt, Gerrit Eichner and Mathias Fasshauer
First of all I want to thank you for opportunity to review this interesting manuscript.
In the presented article, authors try to estimate the influence of alcohol, coffee or tea intake on risk of developing dementia. Introduction is giving enough information about the undertaken problem. Methods are described extensively but I find some information missing. Results are presented very clearly and are easy to follow. Discussion is written logically and based on obtained results but I am missing some straight conclusions.
Overall I find this manuscript very interesting, well-written and important due to undertaken problem and obtained results.
Major concerns:
- I do not quite understand why the authors divided alcoholic beverages only into two groups (wine vs non-wine), I believe that grouping low alcoholic beverages like beer (6%) with strong alcoholic beverages like vodka (40%) is a serious mistake or am I missing an explanation for this? This needs to be better addressed in the manuscript.
- I did not find information about observations included in the analysis of coffee or tea influence on dementia, I wonder if authors included only lifetime alcohol abstinent. Ideally to measure coffee influence there should be a group, which did not drink tea and vice versa, however I understand that this is nearly impossible.
- In discussion paragraph I am missing some conclusion on the basis of obtained results. For example, I see that moderate wine consumption causes “neuroprotective” effect, however non-wine beverages do not induce it. In my opinion this is an effect caused by other substances included in wine (e.g. resveratrol (doi: 10.3892/mmr.2019.10723, 10.1002/biof.1396) but not in other drinks, this is completely not discussed in “Discussion” paragraph.
Minor concerns:
- Table 1. There are no other ethnic groups in table except “White”
Round 2
Reviewer 1 Report
The authors adressed exhaustively all observations and concerns mentioned in the review report. The revised version has reached a mature and convincing shape. Well done!
I have no additional concerns or observations.
Reviewer 2 Report
Authors responded to all my concerns. I have no more remarks. Thank you and congratulations on your work!